# Scale-Adaptive Simulation of Unsteady Cavitation Around a Naca66 Hydrofoil

**Víctor Hidalgo** [1,2,*](), **Xavier Escaler** [3](), **Esteban Valencia** [1], **Xiaoxing Peng** [4], **José Erazo** [2], **Diana Puga** [2] **and Xianwu Luo** [5]

1   Departamento de Ingeniería Mecánica, Escuela Politécnica Nacional, Quito 170525, Ecuador
2   Laboratorio Informática–Mecánica, Escuela Politécnica Nacional, Quito 170525, Ecuador
3   Centre for Industrial Diagnostics and Fluid Dynamics, Universitat Politècnica de Catalunya, 08028 Barcelona, Spain
4   National Key Laboratory of Ship Vibration and Noise, China Ship Scientific Research Center, Wuxi 214082, China
5   State Key Laboratory of Hydro Science & Engineering, Tsinghua University, Beijing 100084, China
*   Correspondence: victor.hidalgo@epn.edu.ec; Tel.: +593-982-491-193

**Abstract:** The present paper focuses on the numerical simulation of unsteady cavitation around a NACA66 hydrofoil to improve the understanding of the cavitation effects on hydraulic machinery. For this aim, the Zwart–Gerber–Belamri cavitation model was updated and uploaded as a library file for OpenFOAM's solvers using C++ language. Furthermore, the hybrid Reynold average Navier–Stokes (RANS)–large eddy simulation (LES) model $k - \omega$ SST scale adaptive simulation (SAS) was implemented as a turbulence model for the present study of scale adaptive simulation. For validation, numerical results were compared with experimental results obtained by Leroux at the Naval Academy Research Institute in France. In order to highlight the benefits in terms of computational consumption and reproduction of the phenomenon the $k - \omega$ SST SAS model was compared against implicit large eddy simulation (ILES). Results show that the cavitation evolution including the maximum vapor length, the detachment and the oscillation frequency were reproduced satisfactorily using $k - \omega$ SST SAS. Moreover, $k - \omega$ SST SAS results predicted a lower total vapor volume on time than ILES, which is related to observed pulses of pressure coefficient, $C_\mathrm{p}$, and those match fairly well with the experimental results. To summarize, the $k - \omega$ SST SAS model predicts with good accuracy unsteady cavitation behavior around hydrofoils and shows improved versatility over the ILES approach.

**Keywords:** $k - \omega$ SST SAS; LES; OpenFOAM; unsteady cavitation

## 1. Introduction

Studies of unsteady cavitating flow around hydrofoils are important to understand the cavitation dynamic behavior and its impact in hydraulic machinery [1,2]. Experimental studies have shown limitations such as: equipment uncertainty, costs and experiments calibration [3]. Therefore, CFD is an alternative way to study the aforementioned phenomenon based on numerical models [4].

One of the main characteristics of cavitating flows around foil or blade is the detachment process of the cavities, which should be related to eddies of the flow field [5]. In previous studies, large eddy simulation (LES) has been used to reproduce the unsteady behavior [6–8]. LES can be applied using explicit or implicit methods to estimate the subgrid tensor and consequently large and small eddies [9]. Based on previous studies of Lu et al. [10], Hidalgo et al. [4] applied implicit large eddy simulation (ILES) for the numerical simulation of unsteady cavitating flows around a plane–convex hydrofoil

and results showed that the main characteristics of the cavitation phenomenon were reproduced fairly well. However, LES models are more complex than RANS, and they have been underutilized to estimate the regular growth of the leading edge cavity at the beginning of the cavitation phenomenon. Considering that point of view, researchers such as Huang et al. [11] and Ji et al. [12] applied the partially averaged Navier–Stokes method (PANS) for the numerical simulation of turbulent cavitating flows based on Girimaji ideas [13]. PANS is a hybrid turbulence model, which conducts the simulation from Reynold average Navier–Stokes (RANS) to direct numerical simulation, DNS. It is based on unresolved total ratios of kinetic energy, $f_k$, and dissipation, $f_\varepsilon$. Nevertheless, those ratios can only be found by a subgrid independence analysis and they are not easily estimated [13,14]. Other authors as Li et al. [15], applied the unsteady RANS (URANS) to achieve the unstable behavior of the cavitation phenomenon. The URANS averages out all turbulent fluctuations and resolves only frequencies far lower than those of the turbulent fluctuations, which are resulting from geometry variations or boundary conditions and typically only reproduce the single vortex shedding frequency [16,17]. Thus, the cavitation shedding and the detached cavity are not fully reproduced in Li's studies [15]. In this context, a hybrid model RANS–LES can be applied to reproduce the attached cavity and the flow separation.

The detached eddy simulation (DES) mixes the computational cost of RANS in the attached boundary layer and of LES in the separation region [18,19]. However, the grid should be highly refined in a region of particular interest, and the simulation depends in a great percentage of the grid mesh as in a similar way to LES. In contrast, Menter and Egorov. [17] proposed a RANS–LES turbulence model called scale adaptive simulation (SAS) which is based on previous studies carried out by Menter [20,21]. The turbulence model is similar to DES, but without an explicit influence of the grid mesh at the RANS mode of the model, and it changes smoothly from a LES model back to steady RANS. Moreover, SAS presents benefits in comparison to URANS due to the spectral content of turbulent flows [17,19]. After all, the use of SAS for numerical simulations of unsteady cavitating flows around hydrofoils has not been widely explored yet, and the model mechanics of the scale–adaptive characteristics in regions as the detached process of cavities is not fully understood [19]. Therefore, the present study is focused on the application of SAS to research unsteady cloud cavitation flows around a NACA66 hydrofoil with experimental validation based on [22].

Moreover, for a better understanding of the SAS model and to capture major trends, ILES has also been carried out for the numerical simulation of unsteady cloud cavitation flows around a NACA66 hydrofoil. Then, results have been compared to show the matching of numerical prediction, and their differences. The comparison is based on the methodology proposed by Edgar et al. [23].

## 2. Description of Numerical Models

### 2.1. Implicit Large Eddy Simulation(Iles)

Filtered equations of continuity and momentum have been used for ILES based on previous studies of Hidalgo et al. [4,6] and Lu et al. [10], which were related to implicit large eddy simulation of unsteady cavitation around hydrofoils as

$$\frac{\partial \rho}{\partial t} + \frac{\partial (\rho \overline{u}_i)}{\partial x_i} = 0, \tag{1}$$

$$\frac{\partial}{\partial t}(\rho \overline{u}_i) + \frac{\partial}{\partial x_j}(\rho \overline{u_i u_j}) = -\frac{\partial \overline{p}}{\partial x_i} + \frac{\partial}{\partial x_j}\left[\rho v \left(\frac{\partial \overline{u}_i}{\partial x_j} + \frac{\partial \overline{u}_j}{\partial x_i}\right)\right], \tag{2}$$

where $\overline{u}$ is the filtered velocity, $\overline{p}$ is the filtered pressure, $t$ is the time, $\rho$ and $v$ are fluid density and kinetic viscosity, respectively, and $_i$ and $_j$ are the subscripts for space coordinates. Moreover, the following considerations have been taken into account:

1.   The product of filtered velocities is $\overline{u_i u_j} = \overline{u}_i \overline{u}_j + \overline{u'_i u'_j}$.

2. The subgrid stress tensor is $\tau'_{ij} = \rho \left( \overline{u_i u_j} - \overline{u}_i \overline{u}_j \right)$.
3. The filtered strain tensor rate is $\overline{S}_{ij} = 0.5 \left( \partial \overline{u}_i / \partial x_j + \partial \overline{u}_j / \partial x_i \right)$
4. The filtered viscous stress tensor is $\overline{\tau}_{ij} = 2 \rho \nu \overline{S}_{ij}$.

Finally Equation (2) is modified and becomes:

$$\frac{\partial}{\partial t} (\rho \overline{u}_i) + \frac{\partial}{\partial x_j} (\rho \overline{u}_i \overline{u}_j) = -\frac{\partial \overline{p}}{\partial x_i} + \frac{\partial \left( \overline{\tau}_{ij} - \tau'_{ij} \right)}{\partial x_j}. \tag{3}$$

According to Lu et al [10] $\tau'_{ij}$ is a nonlinear term, and it is expressed as

$$\tau'_{ij} \equiv \rho \left( \overline{\overline{u}_i \overline{u}_j} + \widetilde{\tau}'_{ij} \right), \tag{4}$$

where $\widetilde{\tau}'_{ij}$ is modeled by using the truncation error of the discretization to obtain an implicit subgrid scale (SGS) for ILES in OpenFOAM [24]. According to Rodi et al. [25], it can be expressed as

$$T_{\text{error}} \cong \frac{\partial \widetilde{\tau}'_{ij}}{\partial x_j}, \tag{5}$$

where $T_{\text{error}}$ is the truncation error of the $\vec{T}$ vector in $i$ direction for a control volume. Thus, ILES can be similar to the classic explicit LES, if the truncation error is close to the SGS action [14]. Adams et al. [26] give more details about the mathematical implementation of a general ILES.

## 2.2. Scale Adaptive Simulation (SAS)

In SAS, the $\tau'_{ij}$ is solved by using the length scale to perform the turbulent viscosity as

$$L_{\nu K} = \kappa \left| \frac{\overline{U}'}{\overline{U}''} \right|, \tag{6}$$

where $\kappa$ is the von Kármán constant and it is usually equal to 0.41 according to Xu et al. [27], $L_{\nu K}$ is a three-dimensional generalization of the boundary layer definition considering a von Kármán length-scale, with

$$\overline{U}' = \sqrt{2 \overline{S}_{ij} \overline{S}_{ij}}, \tag{7}$$

and

$$\overline{U}'' = \sqrt{\frac{\partial^2 \overline{u}_k}{\partial x_i^2} \frac{\partial^2 \overline{u}_k}{\partial x_j^2}}, \tag{8}$$

where $k$ is another subscript for space. In this context, the SAS term is an additional production term in the $\omega$ equation that increases when the flow equations start to unsteady behavior in the $k - \omega$ SST SAS turbulence model [28]. More details of SAS description are given in Menter and Egorov [17].

## 2.3. Zwart–Gerber–Belamri Cavitation Model

The Zwart–Gerber–Belamri Cavitation model (ZGB) was developed to be implemented into the CFX-5 software for the numerical simulation of unsteady cavitation [29]. Considering good results of previous studies [30,31], the ZGB model has been implemented in OpenFOAM version 2.2.x to study unsteady cloud cavitation around hydrofoils [4,14]. In the present study, the ZGB model has been updated and uploaded for OpenFOAM version 4 based on studies of Hidalgo et al. [4]. The vapor volume fraction $\alpha$, the density $\rho$, and the dynamic viscosity $\mu$ of the vapor-water mixture are defined as follows:

$$\alpha = \frac{V_{\text{v}}}{V}, \tag{9}$$

$$\rho = (1 - \alpha)\,\rho_{\mathrm{l}} + \alpha\rho_{\mathrm{v}}, \tag{10}$$

$$\mu = (1 - \alpha)\,\mu_{\mathrm{l}} + \alpha\mu_{\mathrm{v}}, \tag{11}$$

where $V$ is the total volume, $_\mathrm{l}$ and $_\mathrm{v}$ are subscripts for liquid and vapor, respectively. Including $\alpha$ in Equation (1) for the effect of the phase transformation, we have

$$\frac{\partial\,(\alpha\rho_{\mathrm{v}})}{\partial t} + \frac{\partial\,(\alpha\rho_{\mathrm{v}}\bar{u}_i)}{\partial x_i} = \dot{m}^+, \tag{12}$$

$$\frac{\partial\,((1 - \alpha)\rho_{\mathrm{l}})}{\partial t} + \frac{\partial\,((1 - \alpha)\rho_{\mathrm{l}}\bar{u}_i)}{\partial x_i} = \dot{m}^-, \tag{13}$$

where $\dot{m}$ is the inter-phase mass transfer rate per unit volume, and the result of $\dot{m}^+ + \dot{m}^-$ is equal to 0. Finally, the ZGB model indicated in Equation (14) was written in C++ code and implemented in OpenFOAM.

$$\dot{m} = \begin{cases} \dot{m}^+ = & F_{\mathrm{v}}\dfrac{3r_{\mathrm{nuc.}}\,(1-\alpha)\rho_{\mathrm{v}}}{R_{\mathrm{B}}}\sqrt{\dfrac{2}{3}\left(\dfrac{p_{\mathrm{v}}-p}{\rho_{\mathrm{l}}}\right)} & \text{if}\quad p < p_{\mathrm{v}} \\[3mm] \dot{m}^- = & -F_{\mathrm{c}}\dfrac{3\alpha\rho_v}{R_{\mathrm{B}}}\sqrt{\dfrac{2}{3}\left(\dfrac{p-p_{\mathrm{v}}}{\rho_{\mathrm{l}}}\right)} & \text{if}\quad p > p_{\mathrm{v}} \end{cases}, \tag{14}$$

$F_{\mathrm{v}}$ and $F_{\mathrm{c}}$ are the calibration constants for vaporization and condensation, $r_{\mathrm{nuc.}}$ is the nucleation site fraction and $R_{\mathrm{B}}$ is the typical bubble size in water [4]. Furthermore, the model was optimized using $F_{\mathrm{v}} = 300$, $F_{\mathrm{c}} = 0.03$ and $r_{\mathrm{nuc.}} = 5 \times 10^{-6}$, which are based on Morgut et al. [32].

## 3. Hydrofoil Geometry and Computational Domain

The geometry of the NACA66 hydrofoil and the main computational domain are outlined in Figure 1a,b respectively and it is noted that the surface roughness of NACA66 body was considered as a no-slip wall condition. The chord length ($c$) is equal to 0.15 m with an attack angle ($\widehat{AOA}$) of $6°$, which were based on experimental studies carried out by Leroux [22]. According to Ji et al. [33], boundary conditions for inlet and outlet are 5.48 m/s and 20.3 kPa, respectively, and the value of cavitation number ($\sigma$) is 1.2.

A structured grid mesh was obtained using GMSH, which is a free open source software [34]. For that, an exponential criteria close to the boundary layer was applied to improve the CFD performance, and the resulting grid is composed by 801628 cells with 38 divisions on the span-wise direction as indicated in Figure 2a, which is based on the mesh independence analysis carried out by Ji et al. [33] and Yu et al. [35]. The mesh requirements of ILES and SAS have been achieved with a $y^+$ equal to 2 [4,17]. Further mesh analysis was considered using the quality filter of Paraview [14], and results show values between 0.7 and 1, which are acceptable according to previous studies [4,14] as indicated in Figure 2b.

The Yplus, $y^+$, was calculated as

$$y^+ = \frac{u_\tau y}{\nu}, \tag{15}$$

where $u_\tau$ is the friction velocity and $y$ is the distance to the nearest hydrofoil wall. Furthermore, the cavitation number, $\sigma$, was calculated as

$$\sigma = \frac{p_{\mathrm{r}} - p_{\mathrm{v}}}{0.5\rho U_\infty^2}, \tag{16}$$

where $U_\infty^2$ is the undisturbed flow velocity, $p_{\mathrm{r}}$ and $p_{\mathrm{v}}$ are the reference and saturation pressures, respectively.

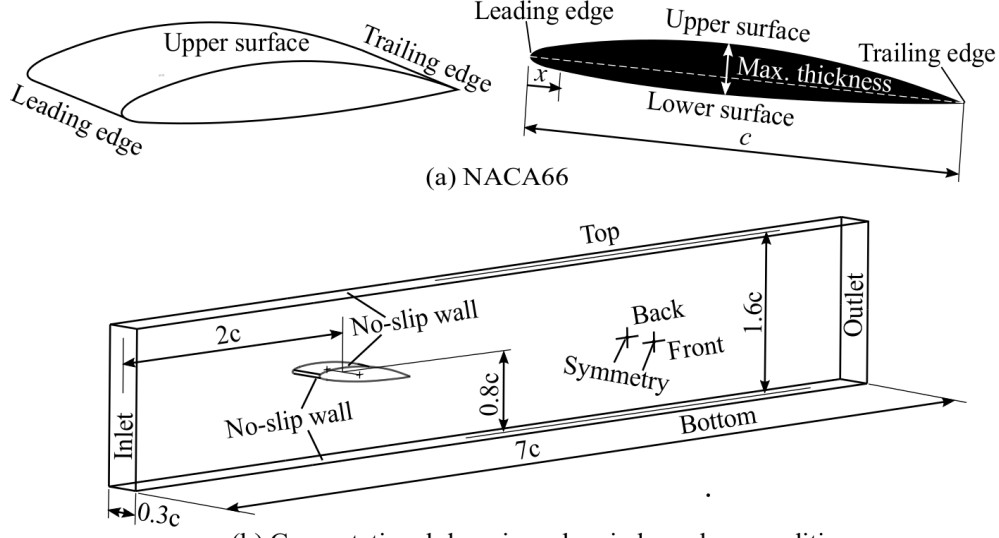

(a) NACA66

(b) Computational domain and main boundary conditions

**Figure 1.** Geometry of NACA66 and computational domain.

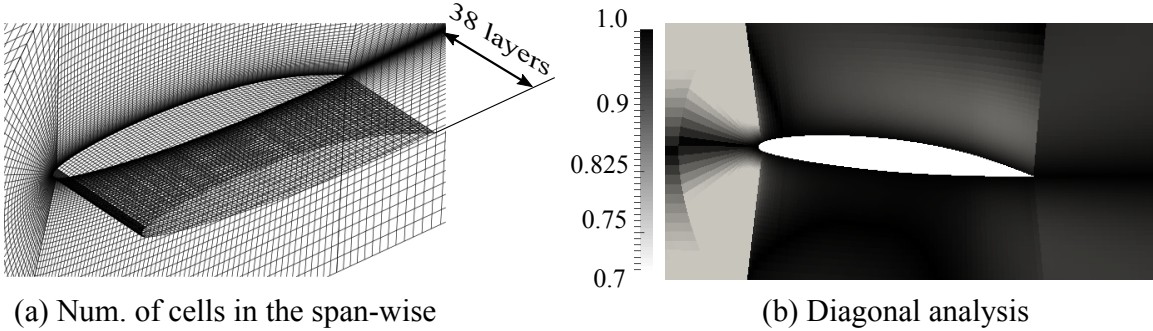

(a) Num. of cells in the span-wise

(b) Diagonal analysis

**Figure 2.** Structured grid mesh around a NACA66.

Numerical simulations were run in parallel using the simple decomposition method of OpenFOAM and OpenMPI. For that, the computational domain was decomposed in eight subdomains to run in eight cores of the Dell Precision 3430 workstation with 64 RAM memory at Laboratorio de Informática–Mecánica of Escuela Politécnica Nacional. Moreover, the mean computational time required to run SAS and ILES simulations were 4 and 6 days, respectively.

## 4. Results and Discussion

For a better understanding of the unsteady cavitation cycles, the total vapor volume, $V_{cav.}$, was estimated using Equation (17), which is based on studies of Ji et al. [30]. The $V_{cav.}$ was divided by the volume of the total computational domain, $V_{domain}$, due to in fact that $V_{cav.}/V_{domain}$ represents a dimensionless volume that is plotted in Figure 3a,

$$V_{cav.} = \sum_{q=1}^{N} [\alpha V]_q ,$$ (17)

where $_q$ is the subscript for each cell and $N$ is the total number of cells in the computational domain. Results show three cavitation cycles from 0 to 0.7 s for both turbulence models. However, the $V_{cav.}/V_{domain}$ of ILES is twice higher than $k - \omega$ SST SAS. This is probably due to the fact that ILES depends on the truncation error of the numerical simulation, which induces false responses in the prediction of $V_{cav.}$. Finally, the fast Fourier transform was applied to find the cavitation frequency and plotted in Figure 3b. The highest pulse is observed at 4.01 Hz in both cases, and it matches the

frequency of the experimental result obtained by Leroux [22]. Moreover, ILES presented additional pulses related to SGS, which may contribute to the large computational resources demanded by ILES.

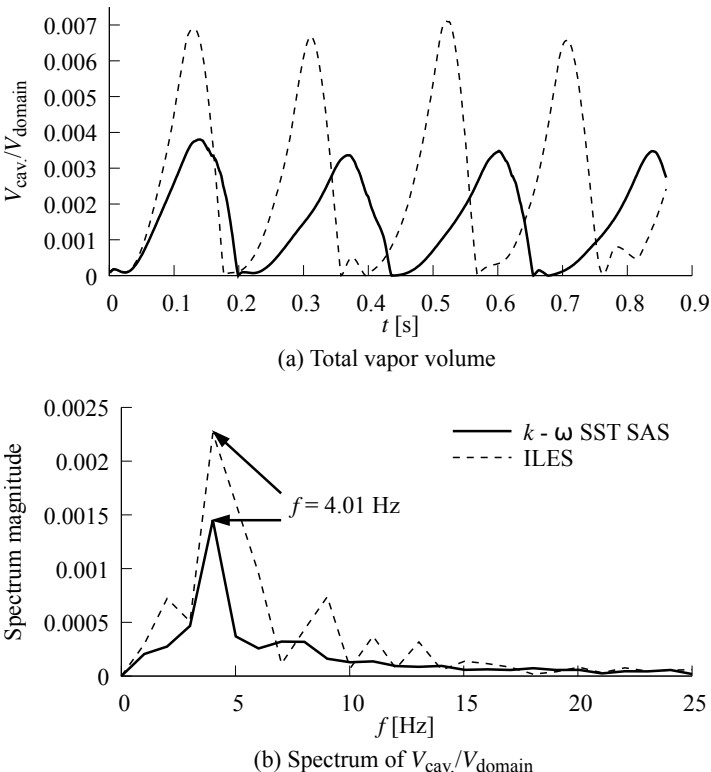

(a) Total vapor volume

(b) Spectrum of $V_{\text{cav.}}/V_{\text{domain}}$

**Figure 3.** The total vapor variation during unsteady cavitation around a NACA66 hydrofoil with $k - \omega$ SST scale adaptive simulation (SAS) and implicit large eddy simulation (ILES).

The third cycle of each case from Figure 3 has been selected and plotted in Figure 4 as function of the dimensionless time, $\xi$, and it was estimated using Equation (18) for a better comparison among cycles, which is based on previous research [4,6].

$$\xi = \frac{t - t_{\text{o}}}{t_{\text{f}} - t_{\text{o}}},\tag{18}$$

where $t$ is time, $_{\text{o}}$ and $_{\text{f}}$ are subscripts for initial and final time respectively. Five instants have been selected in Figure 4 and plotted in Figure 5 to understand the unsteady cavitation cycle.

Figure 5 shows a typical cavitation cycle for numerical and experimental results with the growth and detachment of the leading edge cavity from ① to ③, the development of the re-entrant jet and the collapse of the cloud of cavities from ③ to ⑤. However, the detachment process of cavities is less turbulent for $k - \omega$ SST SAS than ILES at instants ③ and ④ as indicated in Figure 5a,c, respectively. In this context, the cavitation cycle has been reproduced fairly well with $k - \omega$ SST SAS, which is according to previous studies carried out by Ji et al. [30] and Hidalgo et al. [4].

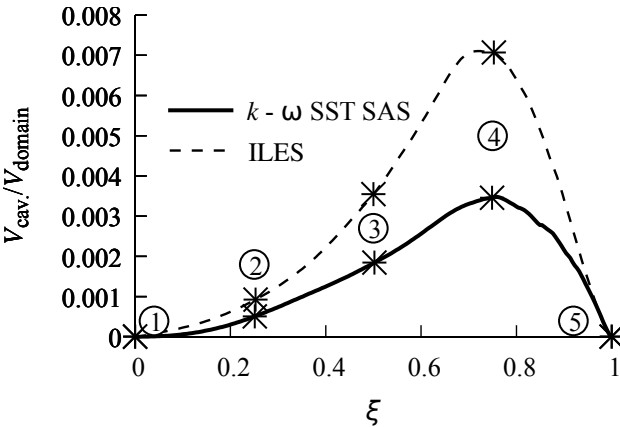

**Figure 4.** Selected instants at the third cycle of both cases.

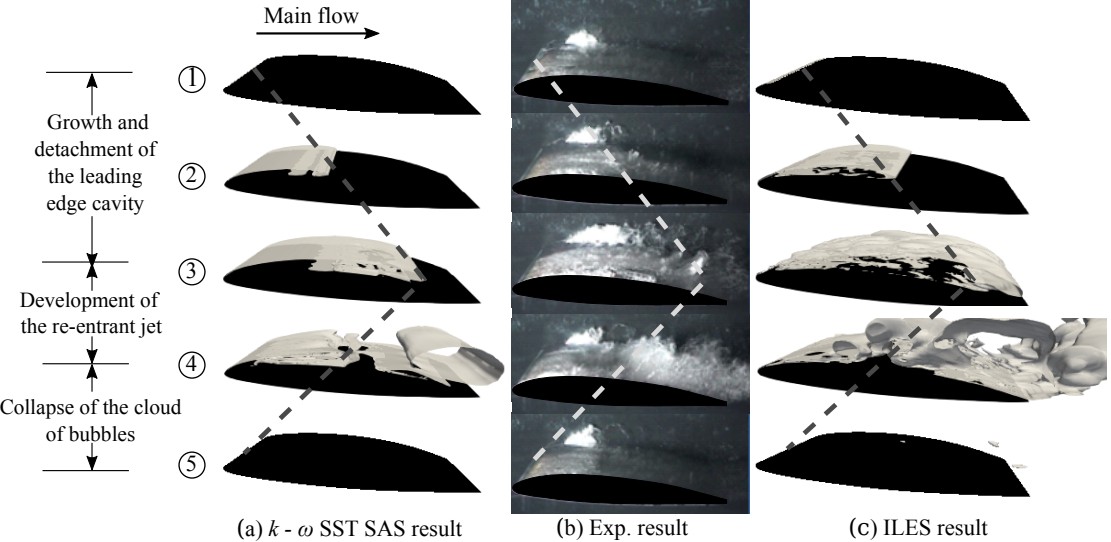

**Figure 5.** Typical cavitation cycle and comparison among results of NACA66 with iso–contour of $\alpha = 0.1$.

Further analysis of the cavity detachment is indicated in Figure 6. Results of $k - \omega$ SST SAS present a more regular detachment process of the cavity than ILES, and they match fairly well experimental results, which were based on Leroux conditions [22], and obtained by Xiaoxing Peng at National Key Laboratory on Ship Vibration and Noise of the China Ship Scientific Research Center. Moreover, it is observed that the growth of the leading edge cavity is regular without any detachment at the initial growth for $k - \omega$ SST SAS. However, the ILES result shows several cavity detachments at the start of the growth of the leading edge cavity, which is related to the SGS and the truncation error [10,24].

For validation of results, the pressure coefficient, $C_p$, has been estimated using Equation (19) for four points, which were located on the hydrofoil upper surface and plotted with bar errors as indicated in Figure 7.

$$C_p = \frac{p_s - p_\infty}{0.5 \rho U_\infty^2}, \tag{19}$$

where $p_s$ is the static pressure.

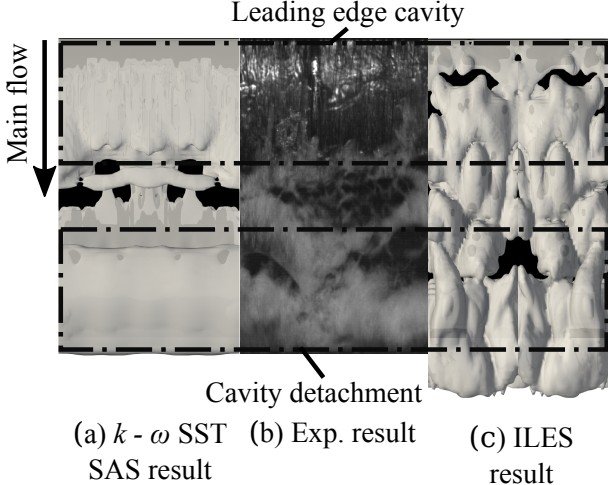

**Figure 6.** Comparison of the leading edge cavity and cavity detachment between numerical simulation results and the experimental result for instant ④ in horizontal plane using iso-contour of $\alpha = 0.1$.

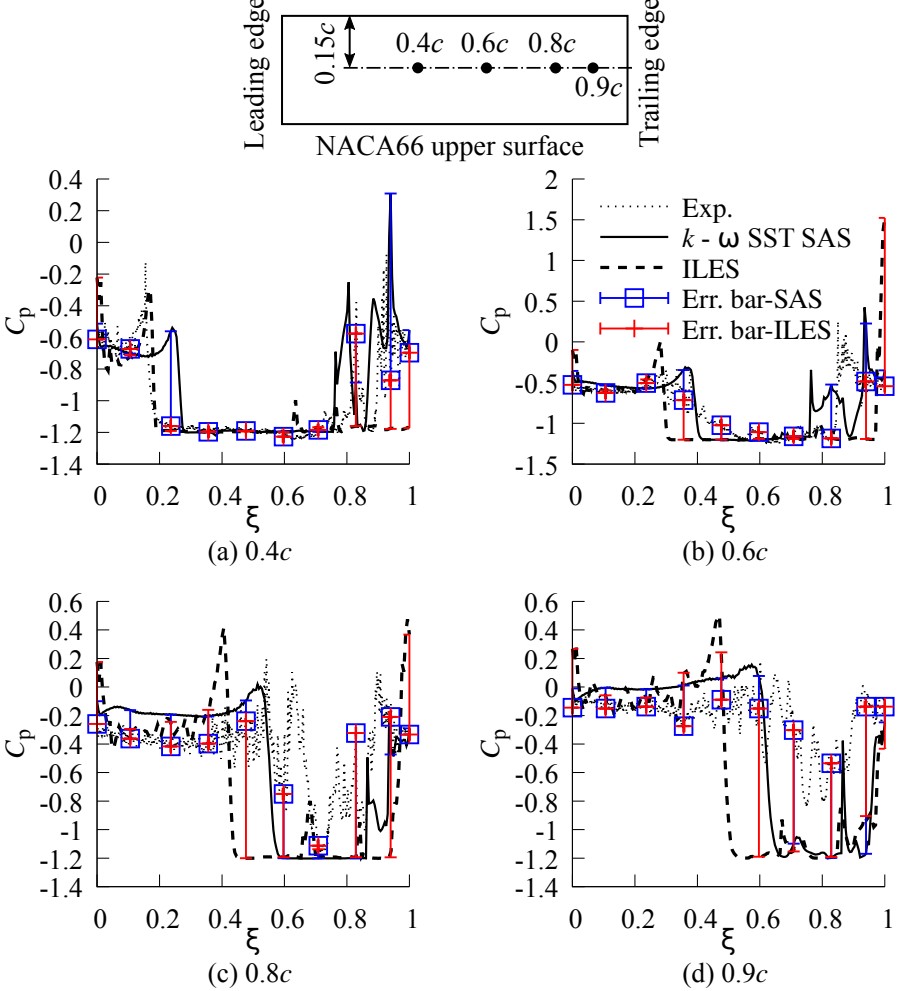

**Figure 7.** Comparison of pressure coefficient, $C_p$, among results.

Results for $k - \omega$ SST SAS, ILES and experiments showed similar trends before $\xi = 0.8$ for points (a) $0.4c$ and (b) $0.6c$, and for the four points before $\xi = 0.4$ due to in fact that the probably detachment process appears at (b) $0.6c$ when $\xi$ is among 0.6 and 1. The $k - \omega$ SST SAS reproduces the trend of experimental results better than ILES from $\xi = 0.4$ to $\xi = 0.6$ as indicated in points (c) $0.8c$ and (d) $0.9c$, which is an advantage to understand the phenomenon of unsteady cavitation and cavitation

detachment against ILES. According to Hidalgo et al. [4], the $C_p$ can be equal to $-\sigma$ when $p_s$ achieves values of $p_v$, which is observed in Figure 7 for $\xi$ values among 0.2 and 0.8. However, the experimental study presents differences at (d) $0.9c$ due to the fact that $C_p$ is higher than $-\sigma$ with more pulses than numerical results, which are related to the cavity collapse process [3]. For that, the $k - \omega$ SST SAS approach matched better the trend of the experimental results than ILES approach.

## 5. Conclusions and Future Works

Numerical simulations of unsteady turbulent cavitation flows around a NACA66 hydrofoil were carried out using OpenFOAM version 4, and the main remarks are:

- The unsteady behavior of the cavitation phenomenon was reproduced using the $k - \omega$ SST SAS and ILES turbulence models. Comparisons among results show that SAS reproduces fairly well the unsteady behavior of the cavitation phenomenon with a cycle frequency of 4.01 Hz, which matches experimental results. The growth of the leading edge cavity was regular, and it was performed with RANS conditions using SAS without showing any detached cavity process at the beginning of the cycle, which is an improvement in comparison to LES results. Therefore, the proposed use of SAS to reproduce unsteady cavitating flows, has been validated as a reliable hybrid turbulence model to be applied in studies of the unsteady cavitation around hydrofoils.
- The Zwart–Gerber–Belamri cavitation model was updated and implemented in OpenFOAM version 4 to simulate liquid–vapor phase changes, and it showed good accuracy against experimental results from Leroux at Naval Academy Research Institute.
- The main contribution of this work was to explore a new turbulence model based on RANS for the study of unsteady cavitation, which presents potential applications for hydraulic machinery design due to low computational demand and high phenomenon reproducibility. Future work will be focused on the implementation of the aforementioned model on optimization routines for hydraulic turbine runners.

**Author Contributions:** Conceptualization, V.H.; methodology, V.H., X.L. and X.E.; software, V.H., J.E. and D.P.; validation, V.H., X.P. and X.E.; formal analysis, V.H., X.E., X.L., X.P. and E.V.; investigation, V.H., X.L. and X.E.; writing—original draft preparation, V.H., X.E., E.V., X.L. and J.E.; writing—review and editing, V.H., E.V., J.E. and D.P.; visualization, V.H.; supervision, V.H.; project administration, V.H.; funding acquisition, V.H. and E.V.

**Funding:** This research was financially supported by Escuela Politécnica Nacional (Project No. PIJ 17-13).

**Acknowledgments:** The authors gratefully acknowledge the financial support provided by Escuela Politécnica Nacional for the development of the project PIJ 17-13. They also thank the support given by the State Key Laboratory of Hydroscience and Engineering, Tsinghua University.

**Conflicts of Interest:** The authors declare no conflict of interest.

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
