# Peer review of "Scale-Adaptive Simulation of Unsteady Cavitation Around a Naca66 Hydrofoil"

_applsci, doi:10.3390/app9183696_

Round 1
Reviewer 1 Report
1- Eq. (1)-(14): They need proper referencing, citation or a support from the literature, otherwise, derivation must be given in the paper with elaboration on the assumption on how the authors did the derivation.
2- Does the model ignore the effect of the surface roughness of the NACA66 body?
3- Fig. 5 needs further elaboration and discussion.
4- A paragraph on the potential application of the developed model for other applications must be added to the paper.
5- Model needs further validation against the already reported data in the literature. Use error bar when comparing the results with the literature clarifying the accuracy of the model.
6- More elaboration on Mesh Independence analysis is required. How the authors come up with the best meshing strategy?
7- Literature is shallow and more relevant works can be reviewed and used in the literature part. Also, the objective of this work must be further highlighted.
8- The abstract and also the conclusion parts need further elaboration.
Author Response
1- Eq. (1)-(14): They need proper referencing, citation or a support from the literature, otherwise, derivation must be given in the paper with elaboration on the assumption on how the authors did the derivation.
The kind suggestion that you made us was considered important to improve the research, so each equation from (1) to (14) was proper referenced based on previous researches.
2- Does the model ignore the effect of the surface roughness of the NACA66 body?
Lines 105 -106 answer your question about roughness.
3- Fig. 5 needs further elaboration and discussion.
The kind suggestion related to Figure 5’s discussion has been implemented in lines 154-159.
4- A paragraph on the potential application of the developed model for other applications must be added to the paper.
The kind suggestion related to potential applications of the development of the model has been implemented in lines 199-203.
5- Model needs further validation against the already reported data in the literature. Use error bar when comparing the results with the literature clarifying the accuracy of the model.
The kind suggestion related to model validation was implemented in Figure 7 and the accuracy clarification is detailed in lines 160-182.
6- More elaboration on Mesh Independence analysis is required. How the authors come up with the best meshing strategy?
The wise opinion about mesh independence analysis was implemented in lines 110-117.
7- Literature is shallow and more relevant works can be reviewed and used in the literature part. Also, the objective of this work must be further highlighted.
Your kind suggestion related to literature review has been taken into account and implemented in lines 209-294. Please send us the literature that you consider is more relevant for this work.
8- The abstract and also the conclusion parts need further elaboration.
The wise opinion about abstract and conclusions was implemented in lines 1-15 and 188-203, respectively.

Reviewer 2 Report
The articel is wrote very good, the structure is clear and systematic, the description of methods is brief but sufficient with relevant references. One third of the citations are self-citations, which is relatively enough, but given the number of authors and references to their results, this is also probably relevant.
97 In the equation (14) should be for m+ in the numerator (pv-p) and than so the square root is not negative .
125 For the readers could be interesting the runtime of simulations and their comparison for ILES and k-ω SST SAS.
136 - 138 The sentence about ILES and its results is a bit strange, short and difficult to understand. This phenomenon and the results of the comparison are interesting and deserve a more detailed analysis, why are the results of the ILES worse that the results of k-ω SST SAS!
177 The conclusions are relatively short and the future works completely missing.
Author Response
Reviewer 2
The article is wrote very good, the structure is clear and systematic, the description of methods is brief but sufficient with relevant references. One third of the citations are self-citations, which is relatively enough, but given the number of authors and references to their results, this is also probably relevant.
97 In the equation (14) should be for m+ in the numerator (p-p) and than so the square root is not negative.
Your kind suggestion related to equation (14) has been implemented in the mentioned equation.
125 For the readers could be interesting the runtime of simulations and their comparison for ILES and k-ω SST SAS.
Your kind suggestion related to runtime was implemented in lines 128-129.
136 - 138 The sentence about ILES and its results is a bit strange, short and difficult to understand. This phenomenon and the results of the comparison are interesting and deserve a more detailed analysis, why are the results of the ILES worse that the results of k-ω SST SAS!
Your wise suggestion related to ILES results was included in lines 141-142.
177 The conclusions are relatively short and the future works completely missing.
Your wise suggestion related to conclusions and future work was included in lines 188-203.

Reviewer 3 Report
The paper presents numerical simulation results of unsteady state cloud cavitation around a hydrofoil, however it falls far short of the standards for publication in an archival journal.
1. There are many typos and grammatical errors in the paper and they should be corrected. Some of them are as below.
· Do not use abbreviations in Abstract and Conclusion sections.
· Abstract section: “…cavitation model was updated and upload…” , and “…RANS-LES model k-w SST SAS…” should be rewritten
· Abstract section: Institute name should be in English rather than French. Other institute names are also changed in English.
· Line 103 on page 4: “The chord length, c, it is equal to … equal to 6o” should be revised. Many other sentences should be rewritten.
· Line 107-108 on page 4 should be rewritten.
· Acknowledgement section: “The authors gratefully acknowledgement the financial support...” should be “The authors gratefully acknowledge the financial support...”
2. Mesh independent study is missing and it must be included in the paper.
3. Novelty and new findings should be reported in abstract and conclusions.
4. Better to write conclusions in bullet form, highlighting the major findings.
5. Nomenclature section is missing and it should be included.
6. The term unsteady cloud cavitation used in Line 1 “unsteady cloud cavitation around a NACA66 hydrofoil” is not explained anywhere in the manuscript as it appears in title also. Explain the significance of cloud cavitation in the current study.
7. Numerical results are validated through experimental study performed previously. Add a separate section for validation and comparison and highlight the details of experimental work.
8. Explain the importance of dimensionless time, given in equation (18).
9. Why have you used Equation (16) for cavitation number in the manuscript on page 5? Have you drawn some physical meaning out of it? Explain it in detail.
10. What’s significance of this ratio “Vcav/Vdomain” on page 6? Explain it.
Author Response
Reviewer 3
1. There are many typos and grammatical errors in the paper and they should be corrected. Some of them are as below.
· Do not use abbreviations in Abstract and Conclusion sections.
Your kind suggestion related to abbreviations in abstract and conclusions sections was considered and included in lines 1-15 and 188-203, respectively.
· Abstract section: “…cavitation model was updated and upload…”, and “…RANS-LES model k-w SST SAS…” should be rewritten
Your kind suggestion related to this particular sentence was restructured and rewritten in lines 4-6.
· Abstract section: Institute name should be in English rather than French. Other institute names are also changed in English.
Your kind suggestion related to the institute name was implemented in lines 7 and 198 despite that this is a proper name.
· Line 103 on page 4: “The chord length, c, it is equal to … equal to 6 o” should be revised. Many other sentences should be rewritten.
The kind suggestion related to this sentence was considered, and the sentence was rewritten in lines 106-107.
· Line 107-108 on page 4 should be rewritten.
The kind suggestion related to this sentence was considered, and the sentence was rewritten in lines 107-109.
· Acknowledgement section: “The authors gratefully acknowledgement the financial support...” should be “The authors gratefully acknowledge the financial support...”
The word was rewritten according to your kind suggestion. Please see line 205.
2. Mesh independent study is missing and it must be included in the paper.
Mesh independence study was carried out in previous research cited in this work and included in line 113.
3. Novelty and new findings should be reported in abstract and conclusions.
The kind suggestion related to novelty and new findings in abstract was included in lines 11-15 and in conclusions section in lines 188-203.
4. Better to write conclusions in bullet form, highlighting the major findings.
Your wise suggestion related to conclusions presentation was included in lines 188-203.
5. Nomenclature section is missing and it should be included.
In the paper format, it is no mandatory to include a nomenclature section, so nomenclature is indicated in the paragraphs related to each equation.
6. The term unsteady cloud cavitation used in Line 1 “unsteady cloud cavitation around a NACA66 hydrofoil” is not explained anywhere in the manuscript as it appears in title also. Explain the significance of cloud cavitation in the current study.
Your wise suggestion related to the term “cloud” was considered, and it was removed from the paper as shown in line 1.
7. Numerical results are validated through experimental study performed previously. Add a separate section for validation and comparison and highlight the details of experimental work.
The paragraph was rewritten according your kind suggestion
8. Explain the importance of dimensionless time, given in equation (18).
The kind suggestion related to dimensionless time was explained in lines 146-147.
9. Why have you used Equation (16) for cavitation number in the manuscript on page 5? Have you drawn some physical meaning out of it? Explain it in detail.
The meaning of the Equation (16) is related to previous papers, and it is a common term used for unsteady cavitation and it was explained in lines 178 – 182 too.
10. What’s significance of this ratio “Vcav/Vdomain” on page 6? Explain it.
The kind suggestion related to the significance of “Vcav/Vdomain” ratio was included in lines 133-134.

Reviewer 4 Report
This paper is very interesting but from my point of view it should be improve to be published.
First of all the models could be detailed a little more and more important the explanation why they have been choose should ne explained. Some validations results could be also be added.
The test case presented seems to be referred to Ji et al. (32) but I have not seen any comparisons to results obtained in this article. About the test case only one is a little to few especially with such discrepancies between numerics an measurements.
Speaking about the comparisons between numerical simulation and experiments and I think that more works must be done. Figure 5 is interesting but must be more detailed (more views and some zooms) in order to have a clear understanding of what is going on and not only a bare felling.
Figure 7 should be clarify because too packed to be clear and understandable. The results shown in this figure demonstrate such huge discrepancies with measurements (in both ILES and RANS) that is impossible to conclude anything. Some deeper investigation must be done about that.
Author Response
Reviewer 4
This paper is very interesting but from my point of view it should be improve to be published.
First of all the models could be detailed a little more and more important the explanation why they have been choose should ne explained. Some validations results could be also be added.
The kind suggestion related models detailing was included in section 2.
The test case presented seems to be referred to Ji et al. (32) but I have not seen any comparisons to results obtained in this article. About the test case only one is a little to few especially with such discrepancies between numerics an measurements.
A discussion paragraph was included in the paper.
Speaking about the comparisons between numerical simulation and experiments and I think that more works must be done. Figure 5 is interesting but must be more detailed (more views and some zooms) in order to have a clear understanding of what is going on and not only a bare felling.
The kind suggestion related to Figure 5’s discussion has been implemented in lines 154-159.
Figure 7 should be clarify because too packed to be clear and understandable. The results shown in this figure demonstrate such huge discrepancies with measurements (in both ILES and RANS) that is impossible to conclude anything. Some deeper investigation must be done about that.
The kind suggestion related to model validation was implemented in Figure 7 and the accuracy clarification is detailed in lines 160-182.

Round 2
Reviewer 1 Report
Accept
Reviewer 4 Report
From my points of view the authors have answered to my requests.